# [RE] Bad Seeds: Evaluating Lexical Methods for Bias Measurement

## Reproducibility Summary

**Scope of Reproducibility**

In this work we verify the results of *Bad Seeds: Evaluating Lexical Methods for Bias Measurement* (Antoniak and Mimno, 2021). We replicate the experiments conducted and verify the main claims made in the original paper: (1) Bias measurements depend on seeds and models. (2) Shuffled seed pairs can result in a significant different bias subspace compared to ordered seed pairs. (3) Set similarity is negatively correlated with the explained variance of the first PCA component in the seed pairings subspace.

**Methodology**

We used skip-gram with negative sampling to train word2vec models with the same hyperparameters and data. We implemented code for the experiments using the resulting word embeddings and the seed sets provided by the authors.

**Results**

Overall, only one claim was reproduced. We reproduced the claim that bias measurements is dependent on the choice of seed set. We were not able to adequately reproduce the claims that shuffled pairs of seed sets generally result in less clearly defined correlation and that for pair of seed sets set similarity is negatively correlated with the explained variance of the first principal component.

**What was easy**

The paper is easy to follow. The data was publicly available. Authors replied frequently providing details about the parameters and preprocessing steps. Also authors were open for the discussion regarding seeds on the Github repository of the project.

**What was difficult**

In certain cases, the gathered seed sets json file contained errors. Specifically: 'daughters' was misspelled as 'daughers' (has now been updated). 'ma', 'am' was used as two words instead of one word "ma'am".
The seed words for figure 4 as stated in the appendix are not the same as the one in the image itself. Table 2 from the original paper was also difficult to reproduce as the preprocessing according to the authors' description gave close but not equal results for the NYT dataset and significantly different results for two other datasets. Reproduced numbers are presented in table 3.3. Because of the time constraints, 20 bootstrapped launches were not conducted.

**Communication with original authors**

Contact was made with the original authors on multiple occasions to ask for clarification questions regarding the implementation of the experiments.

# 1 Introduction

Using local context in datasets to generate word embeddings for NLP problems carries the risk of incorporating the original bias of the dataset into the generated model. Techniques to combat such biases requires the measurement of the bias encoded in a model (Bolukbasi et al., 2016). Almost all bias measurement methods rely on lexicons of seed terms to specify stereotypes, while the rationale for choosing specific seeds is often unclear.

In this work we verify the results of *Bad Seeds: Evaluating Lexical Methods for Bias Measurement* (Antoniak and Mimno, 2021). We replicate the experiments conducted and attempt to verify the main claims made in the original paper.

# 2 Scope of reproducibility

The original paper collected many different seeds (gathered and generated ones) for bias measurement and provided various ways these seeds were originally selected. Then the paper conducted different experiments showing how different choices can result in different bias measurements. We focus on replicating these experiments, and addressing the following claims:

- Claim 1: Bias measurements depend on seeds and models. The paper shows this by calculating similarity to the *Unpleasantness* vector using different seeds and different embedding models. We verify this by recreating this experiment and showing different seeds and models result in different bias measurements.

- Claim 2: Shuffled pairs of seed sets generally result in less clearly defined correlation in bias subspace than unshuffled pairs of seed sets.

- Claim 3: Identifying bias is less effective when set pairs are similar. This claim is verified by generating seed pairs and plotting set similarity against the explained variance of the first principal component in the bias subspace.

Finally we explore if BERT embeddings results in similar results compared to word2vec models.

# 3 Methodology

## 3.1 Bias Measurement Algorithms

### 3.1.1 PCA

Principal component analysis can be used to measure the variability in the difference vectors between pair of word vectors. The vector that represents this difference the best is used to generate the bias subspace. The PCA algorithm is applied in the following manner: for each pair of seed sets, the mean vector of their two embedding vectors is calculated. The half vectors, which are calculated by subtracting the embedding vectors from the mean, are added to a list. These half vectors are used as the columns in the input matrix of the PCA algorithm. (Bolukbasi et al., 2016). We can further calculate the explained variance ratio of resulting principal components to determine how much variance is explained by the first few principal components.

### 3.1.2 Word Embedding Association Test (WEAT)

To quantify bias, WEAT can be used to find which seed set is more associated to given attribute words (Caliskan et al., 2017). The WEAT score between sets $\mathcal{X}$ and $\mathcal{Y}$, and sets of attributes, $\mathcal{A}$ and $\mathcal{B}$, is defined as

$$s(\mathcal{X}, \mathcal{Y}, \mathcal{A}, \mathcal{B}) = \sum_{x \in \mathcal{X}} s(x, \mathcal{A}, \mathcal{B}) - \sum_{y \in \mathcal{Y}} s(y, \mathcal{A}, \mathcal{B})$$

where $s(w, \mathcal{A}, \mathcal{B})$ is equal to the difference in average cosine similarities between query $w$ and each term in $\mathcal{A}$ and $\mathcal{B}$. A WEAT subspace is created by calculating the vector between the average embeddings of two seed sets. This subspace is used to calculate coherence between pair of seeds. A seed pair is said to have a high coherence when the seeds are

highly separated. Coherence is calculated by ranking the mean ranks of every word in the seeds by the cosine similarity of the bias subspace:

$$\text{Coherence}(\mathcal{X}, \mathcal{Y}) = |\overline{R_\mathcal{X}} - \overline{R_\mathcal{Y}}|$$

where $\overline{R_\mathcal{X}}$ and $\overline{R_\mathcal{Y}}$ are the mean ranks of the seed sets in the bias subspace. These two values are normalized to be between 0 and 1.

## 3.2 Datasets

For reproduction purposes, we used the links to the datasets mentioned in the paper. NYT was easily found on kag. The original link to the WikiText-103 didn't work and a word-level dataset from wik was taken. Goodreads reviews worked with the link provided.

## 3.3 Preprocessing

Most of the details of the preprocessing procedure were found in the paper. However, some steps became clear only after reaching authors who were open to discussion.

The final preprocessing pipeline looked as follows:

- splitting of each dataset into logical documents:
    - articles separated by newline symbols and URLs in NYT;
    - articles separated by an atricle name inside "=" signs in WikiText;
    - separate json files in Goodreads;
- lowercasing;
- removing non-alphanumeric characters;
- splitting on whitespace;
- removing words that occurr fewer than 10 times in the training dataset (via gensim word2vec model min_count argument).

Apart from this, it was suggested to apply additional cleaning steps for WikiText:

- lists removal;
- HTML errors removal;
- math removal;
- code removal.

No details regarding this were found. Moreover, we were not able to recove LaTeXexpressions and HTML tags via regular expressions. So the question about WikiText preprocessing remains open.

The datasets were preprocessed and summary statistics for them were calculated 3.3. It turned out that for WikiText there is a decrease in the vocabulary size and the mean document length compared to the original paper. For Goodreads, there is a difference in the vocabulary size, potentially caused by preprocessing nuances.

| Team | Total Documents | Total Words | Vocabulary Size | Mean Document Length |
|---|---|---|---|---|
| NYT | 8,888 | 7,210,433 | 136,404 | 811 |
| WikiText | 28,470 | 86,397,984 | 223,631 | 3,034 |
| Goodreads (Romance) | 197,000 | 24,116,941 | 291,623 | 122 |
| Goodreads (History/Biography) | 136,000 | 14,001,917 | 216,067 | 103 |

Table 2: Reproduced summary statistics for our datasets

## 3.4 Seeds

Algorithmic bias can be observed in machine learning models when terms associated with a particular gender, ethnicity, political party or other such grouping are treated differently by the model than the average term. To quantify how much

bias is present in a model, seed terms are required that have a strong correlation with a particular bias dimension that we wish to represent. A set of these terms, called a seed set, together form a reference for a particular bias group. A seed set can be used to quantify how much bias is present in a model for a bias group, by looking at the difference in outputs between the seed terms and all other terms. Clearly, it is important that the right seed terms are grouped to measure bias for a particular bias group, otherwise, the bias quantification of a model might be flawed.

### 3.4.1 Gathered seeds

In the original paper, the authors use a collection of seed sets gathered from 18 'highly-cited papers'. We use the same seed sets as used in the experiments for the original paper (Bolukbasi et al., 2016) (Kozlowski et al., 2019) (Garg et al., 2018) (Caliskan et al., 2017) (Manzini et al., 2019) (Zhao et al., 2018) (Hoyle et al., 2019). Like the original work, we use unigram seeds and omit words that were not present in the training set.

### 3.4.2 Generated seeds

Using generated seed sets is a way to get a large number of seed sets to use for experiments. To find suitable seed sets, we want to find words that are close together in the embedding space. We also want to control for POS (obtained with spaCy *en_core_web_sm* model is used) and frequency of the words in the dataset. The way we generate seed sets is by first selecting a random noun in the vocabulary with a probability proportional to the frequency in the dataset. We check if a word is a noun or not using the *spacy* library. Then we take the four nearest nouns of this word ranked by cosine similarity

### 3.5 Experimental Setup

In this section, we will provide steps to replicate the results obtained by the original authors. We found that various steps were not made explicit or used different seed sets as documented. We provide details on the steps we have taken to replicate the experiments.

The codebase [1] consist of two notebooks, one for training the models and one for the experiments and graph generation.

### 3.5.1 Models

Different word embeddings have varying amounts of bias embedded inside the models. To verify the claims made regarding the various effects of seeds on bias measurements, we limit the scope to the word embeddings used in the original paper. For each dataset, we train a word2vec model using skip-gram with negative sampling (SGNS). Apart from that, outside the scope of the original paper we tried using BERT-generated embeddings as well.

The word2vec models for Goodreads, NYT and WikiText are trained in *SGNS_train.ipynb* using *preprocessing.py* script. The notebook consists of preprocessing of the original dataset, the training of the word2vec model and saving the model. All SGNS model parameters were used with default values, namely *5 training epochs, 5 negative samples per each positive, vector size of 100, window size of 5, word frequency not lower than 10*.

Script *bert_embeddings.py* was used to obtain pre-trained embeddings from a widely used *bert-base-uncased* model. Devlin et al. (2018)

### 3.5.2 Experiments

The experiments of the original paper are reproduced in *Paper reproduction.ipynb*. To address claim 1, we replicate figure 2 of the original paper to highlight how much different seeds in the same category affect the similarity to a certain vector in the embedding space. This is done by calculating for both the word embedding using the romance Goodreads reviews and the history + biography Goodreads reviews the cosine similarity of the words in various seed sets with the word *Unpleasant* in the same word embedding. Note that we do not use the word *Unpleasantness* as was used in the original paper, because that word is not contained in the embeddings generated with the Goodreads dataset.

---

[1]The trained models and files for replication are available in an anonymized format at https://anonymous.4open.science/r/xxxxxx-3788

To address claim 2, we reproduce figure 3 of the original paper. Principal component analysis is used to calculate the explained variance of the first ten components of different pairs of seed sets. For each pair of seed set, both the original order is used and a pairing where the seed pairings are shuffled. The model used is trained on the NYT dataset, but we also replicate the same figure for the BERT embeddings. We also reproduce figure 4, which ranks word vectors of different shuffled, unshuffled and random seed pairs by cosine similarity to the first principal component.

To address claim 3, we reproduce table 4 from the original paper by generating 300 seed set pairs using the earlier described method of generating seed sets. We calculate the coherence of each pair using the WEAT bias subspace. We also calculate the coherence for a select amount of gathered seed sets.

The final figure reproduced for addressing claim 3 is created by generating 600 seed pairs. The set similarity for each seed pair is calculated using the cosine similarity between the set mean vectors. For each seed pair, the explained variance of the first PCA component is also calculated. The explained variance is plotted against the set similarity, and similarily to the original paper, a trendline is calculated and plotted over the data.

### 3.6 Computational requirements

Preprocessing, modelling, and visualisation were performed on a personal laptop with 32 GB RAM, 8 CPUs, and 512 GB SSD. Uncompressed datasets took from 45 MB to 4 GB of memory and it is not a challenge to keep them without extra computation services. Datasets were loaded into memory and processed without any batching. Word2vec model was trained using all CPU cores. Plots for BERT embeddings were obtained on Google Colab via GPU.

### 3.7 Results

The figures and tables of the original paper are reproduced in this section. At the hand of the figures and tables, we will go over each claim and state whether we can verify the claim or not.

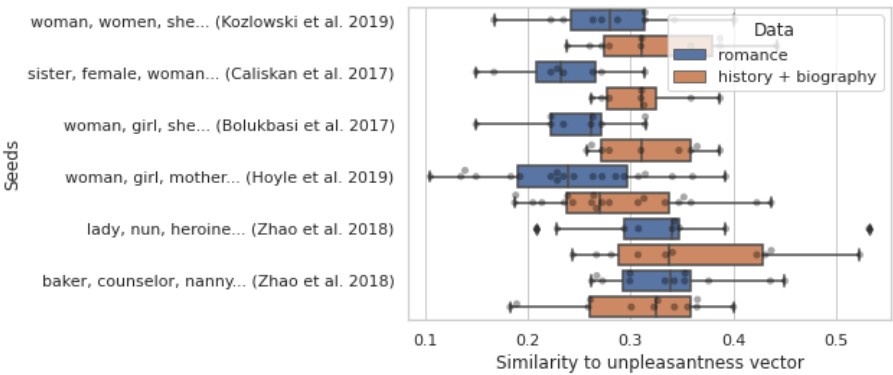

Figure 1: Cosine similarity is plotted for words in different seed sets containing *female* words and the *unpleassant* vector. The word embeddings used are created using both the romance and history+biography Goodreads dataset. The results show that different seed sets result in different bias measurements.

Claim 1: As shown in figure 1, different seed sets result in different bias measurement, even when all seed sets contain the same subject it tries to measure bias for. We can also see that the type of data used to generate the word embeddings can alter the bias measurement. With the replication of this image we can verify claim 1.

Claim 2: In table 1 the cosine similarities for ordered gender pairs indentifies female and male gendered words slightly better than shuffled pairs. Figure 3 show that different pairs of seed sets have a different explained variance distribution for its principal components. While the original paper has a lower explained variance for the first principal components for the gender pairs seed set and a higher explained variance for the first principal components for the other two pairs, we have found a higher explained variance for the first principal component for all pairs of seed sets. Because the results in table 1 suggest the ordered gender pairs only slightly increase the ability to identify gendered words and the results in figure 3 does not suggest that ordered pairs can better define correlation in bias subspace. We therefore cannot verify claim 2.

| (a) Gender Pairs | | (b) Random Pairs | | (c) Random Pairs | | (d) Shuffled Gender Pairs | |
|---|---|---|---|---|---|---|---|
| herself | 0.415 | incentive | 0.277 | lily | 0.36 | herself | 0.395 |
| she | 0.393 | setback | 0.248 | theirs | 0.34 | she | 0.393 |
| female | 0.354 | | | fari | 0.172 | girl | 0.367 |
| her | 0.348 | | | meet | 0.108 | her | 0.339 |
| daughter | 0.292 | | | canoe | -0.043 | daughter | 0.297 |
| boy | -0.165 | | | bilingual | -0.126 | his | -0.175 |
| man | -0.176 | likelihood | -0.106 | brush | -0.202 | himself | -0.191 |
| himself | -0.176 | tales | -0.162 | dictates | -0.304 | son | -0.207 |
| son | -0.245 | hood | -0.567 | longest | -0.424 | male | -0.255 |
| he | -0.260 | danced | -0.682 | julianna | -0.556 | he | -0.264 |

Table 1: Ranking word vectors by cosine similarity with respect to the PCA subspace. The top 5 and bottom 5 words ranked by cosine similarity are plotted, unless certain words are not found in the model trained on the NYT dataset, in which case fewer words are shown. (a) contains gender pairs, (b) and (c) contains random pairs and (d) contains shuffled gender pairs.

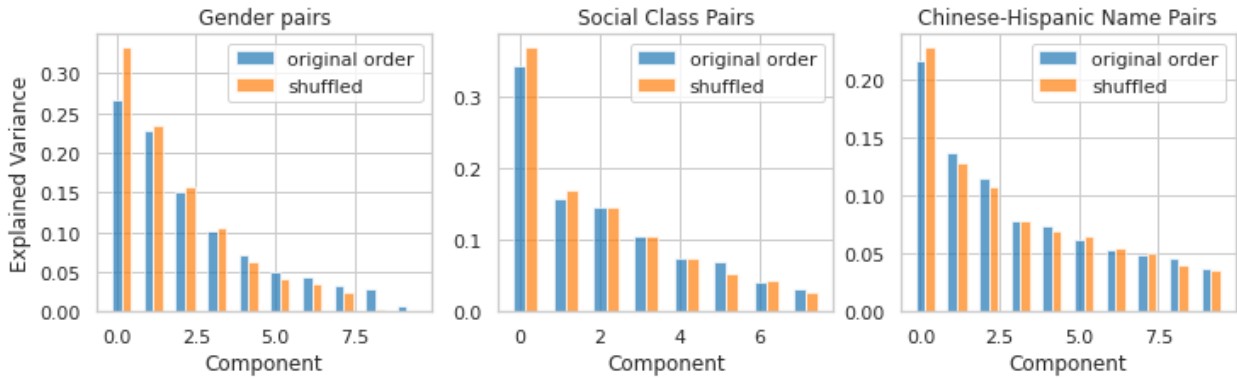

Figure 2: Using PCA, the explained variance for the first 10 principal components are plotted for 3 different pairs of seed sets. The pairs of seed sets are shuffled and the principal components for these new pairs are shown as well. The word embedding used are trained using the New York Times dataset.

Claim 3: For the third claim we first show how different pairs of seed sets have different coherence values. In table 2, different pairs of seed sets, both generated and gathered, are shown. Seed sets with overlapping words or similiar meanings are usually less coherent than more distinct pairs of seed sets. In figure 3, we can not see a significant correlation between the set similarity and the explained variance of the first principal component. We therefore cannot verify the claim made in the original paper.

In figure 4 we show the explained variance for the first 10 principle components using the BERT embedding. BERT shows a very high explained variance for the first principle component regardless of pairs of seed sets or whether it is shuffled or not.

## 4 Discussion

Overall, the main quantitative result of the authors that seed sets contain biases is affirmed by our results, and the choice on which seed set to use to measure bias is an important decission. We were not able to verify the claim made that suggest ordered pairs of seed sets identifies bias more clearly than unordered pairs of seed sets. We were also not able to verify the claim that suggests set similarity to be strongly negatively correlated to the explained variance of the first principal component. The reason we were not able to verify the results might be because of various reasons. The trained models might be different because of slightly different datasets or preprocessing implementation. For figure 3 specifically, the variation might be because of a different way of generating seed sets.

Finally, using embeddings from modern NLP models might show bias in a different way compared to word2vec.

| Coherence | Generated Seed Set A | Generated Seed Set B |
|---|---|---|
| 1.000 | years months decades weeks decade | guards guard commandos gunmen soldiers |
| 1.000 | children parents families kids mothers | front rear porch lobby perimeter |
| 1.000 | statement spokeswoman letter telephone s | market sector volatility swaps markets |
| ... | | |
| 0.320 | blip hassle softness proverbial gloss | eyes lips sunglasses hips ears |
| 0.280 | case dismissal hague juror prosecution | court courts judge judges hague |

| Coherence | Gathered Seed Set A | Gathered Seed Set B |
|---|---|---|
| 0.998 | ASIAN: asian asian asian asia china asia | CAUCASIAN: caucasian caucasian white america |
| 0.997 | FEMALE: sister mother aunt grandmother | MALE 2: brother father uncle grandfather |
| 0.886 | CAREER: executive management professional | FAMILY: home parents children family cousins |
| 0.611 | FEMALE: countrywoman, sororal, witches | MALE: countryman fraternal wizards manservant |
| 0.202 | NAMES ASIAN: cho wong tang huang chu chu | NAMES CHINESE: chung liu wong huang ng |
| 0.049 | NAMES BLACK: harris robinson howard | NAMES WHITE: harris nelson robinson |

Table 2: Using the WEAT subspace, the coherence is calculated for each pair of seed sets. The model used was trained using the NYT dataset. For the generated seed sets, 600 pairs were calculated of which the 3 with the highest and 2 with the lowest coherence were shown. We also show the coherence for a select amount of gathered seed sets.

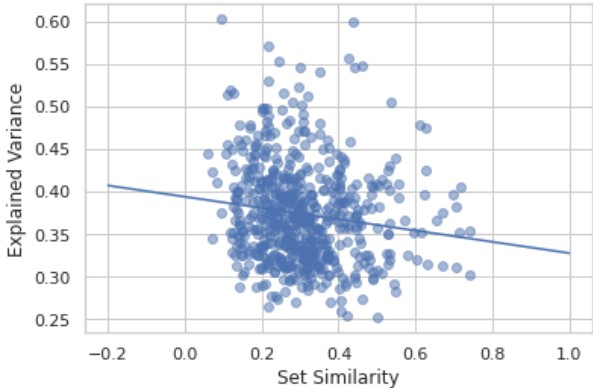

Figure 3: 600 random seed pairs are generated, after which the explained variance of the first principal component and set similarity of each pair are calculated. The WikiText dataset is used to train the word embeddings. A trendline is shown, however, there does not seem to be a significant correlation between the set similarity and explained variance.

### 4.1 What was easy

The paper was relatively easy to follow, and the math was quite straightforward.
As soon as we had determined the correct preprocessing steps after having contacted the original authors, implementing the experiments from the original paper was not too challenging as the data was publicly available.

Another factor that made replicating the original paper easier, is that many points could be clarified by the authors, who were kind enough to respond to our questions.

### 4.2 What was difficult

The code for the original implementation was not available. While implementing the experiments was not too difficult, it was challenging to figure out the subtleties of the data preprocessing steps without having a code example, and in some cases we had to resort to educated guesses. Some of these questions about implementation were cleared up after contact with the authors.

While it was easy to find the datasets online, the references for some datasets were outdated. In the references section of this work we detail the updated sources for the data that was used in this replication.

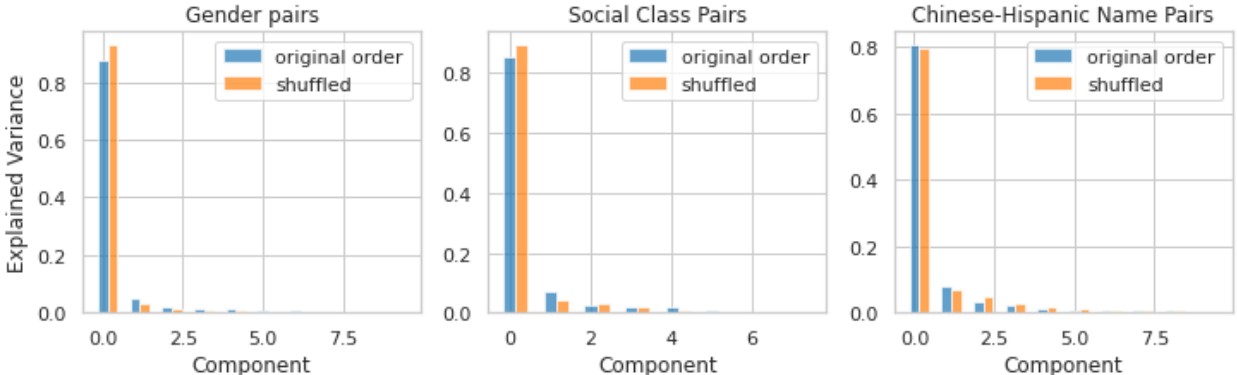

Figure 4: Using PCA, the explained variance for the first 10 principal components are plotted for 3 different pairs of seed sets. The pairs of seed sets are shuffled and the principal components for these new pairs are shown as well. The word embedding used are from a pretrained BERT word embedding with a vector size of 768.

According to the original paper, the WikiText-103 dataset was filtered for mathematical equations. No details were provided on how this was done, and due to the varying format of the equations found in the dataset, it was very challenging to filter them out. It is unclear whether the authors also proceeded using relatively 'dirty' data, or whether they used alternative methods to remove the mathematical equations. Table 2 from the original paper was also difficult to reproduce as the preprocessing according to authors gave close but not equal results for the NYT dataset and significantly different results for the WikiText-103 dataset. For the Goodreads dataset, the set of documents was not fixed but rather randomly sampled from a larger set using an unknown implementation, which made the comparison much more difficult.

In certain cases, the gathered seed sets json file contained errors. Specifically: 'daughters' was misspelt as 'daughers', and 'ma', 'am' was used as two words instead of one word "ma'am". We have notified the authors about this and the seed sets have since been updated.

The seed words for Figure 4 as stated in the appendix of the original paper are not the same as the ones in the image itself, which caused some confusion in replicating the figure.

### 4.3 Communication with original authors

Communication with the original authors played a major role in the reproducibility of the original paper. After receiving details on the pre-processing steps and clarification of some sentences in the original paper, our results came a lot closer to those of the original authors. We have included these details in this paper to be used for future research.

### 4.4 Ablation studies

Ablation studies were not necessary for this paper, as most of the results were quite straightforward to derive, and no multi-faceted methods were used.

### 4.5 Recommendations and future work

Having an original implementation available publicly is invaluable for making research reproducible. If the codebase for research is available, many uncertainties can be immediately cleared up and a major source of uncertainty for reproduction can be eliminated. Furthermore, having the code available would enable other researchers to easily adapt the code to new problems or perform new experiments. The time saved by not having to recreate an implementation could be spent on novel research, enabling the scientific community evaluate and expand on papers more quickly.

Besides this general recommendation, it would be interesting to reproduce this particular research with a more recent word embedding, such as those generated by Transformer-based models.

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
