# OpenReview forum: "[RE] Bad Seeds: Evaluating Lexical Methods for Bias Measurement"
_ML_Reproducibility_Challenge/2021/Fall — Reject_

### Official Review · Reviewer_4Ht7 · 2022-03-17
**[RE] Bad Seeds: Evaluating Lexical Methods for Bias Measurement**

**Rating:** 6
**Confidence:** 5

**Review:**

The authors propose a replication of Antoniak and Mimno's experiment. The source code is freely available, e.g. via the anonymous.4open.science platform. In order to respect the reference article, they have studied 4 datasets initially present in the original study (New York Times, WikiText-103, Goodreads(History/Biography), and Goodreads(romance)). Two metrics were used, namely (WEAT and PCA), to respect and compare the source work. The authors explore if BERT embeddings result in similar results compared to word2vec models.

* The readme is very uninformative. It should be completed with more details.

---

### Official Review · Reviewer_Hhy7 · 2022-03-24
**Review of "[RE] Bad Seeds: Evaluating Lexical Methods for Bias Measurement"**

**Rating:** 4
**Confidence:** 4

**Review:**

**Reproducibility Summary:**
This report tries to reproduce the results presented in the paper titled Bad Seeds: Evaluating Lexical Methods for Bias Measurement (Antoniak and Mimno, 2021). It has a proper first page for the summary of their work. Scope of reproducibility part in this page, the authors pick to use "verify" word to show the purpose of trying to reproduce the results. It sounds like they completely achieve to reproduce all related results, but in the next parts, it is figured out they could not.

**Scope of reproducibility:**
The authors clearly state the scope of reproducibility by picking 3 experiments done in the original work.
- Seed set dependency of bias measurements.
- The effect of shuffling on the bias.
- The correlation between seed set subspace and set similarity of seeds.

Only the first claim can be reproduced by this work. Further discussion about *illuminating* the potential risks posed by seed sets for the bias, which is presented in the original work, could have been included in their discussion. The authors mention that they could not conduct the experiments on repeating the same process across 20 bootstrapped samples of each dataset, due to the time constraint.

**Code:**
Code is implemented from scratch and submitted with an anonymous repository. However, the submitted code lacks the readable documentation to re-run it or how to use it in detail. Anyone using this code should read the code itself, not documentation, to understand the process and further improve it.

Also, the preprocessing of the datasets could not be done exactly the same as the original paper, which may lead to being ineligible to reproduce some parts of the experiments.

**Communication with original authors:**
The report mentions the authors communicate with the original authors to clarify the missing parts.

**Hyperparameter Search:**
The authors do not conduct any experiments for the hyperparameter search, in addition to the main experiments.

**Ablation Study:**
In section 4.4, the authors state that ablation studies were not necessary for this paper, as most of the results were quite straightforward to derive, and no multi-faceted methods were used. However, the reviewer believes that there could be different ablation studies for the original work. For example, the bias measurement can be observed by using different seed sets and a vector for different concepts other than **unpleasantness**. The effect of shuffling can be shown for social class pairs or Chinese-Hispanic pairs.

**Discussion on results:**
The easy and difficult parts of this reproduction study are well-defined by the authors. Some missing but important parts are given in the report where the authors should resolve to complete the preprocessing or acquire the dataset. The possible reasons for not being able to reproduce the study should be more detailed in the discussion part. The effort on being able to achieve reproduction could be expressed in a more elegant way.

**Results beyond the paper:**
The authors mention that they explore if BERT embeddings result in similar when compared to the results of word2vec models. However, in the report, the results are presented with a very limited description of the figure and there is no discussion on why BERT embeddings have a very high explained variance for the first principle component regardless of pairs of seed sets or whether it is shuffled or not. This could be a very helpful discussion for the ones who want to use BERT embeddings for investigating the bias measurement in further studies.

**Overall organization and clarity**
Figures can be placed to the closest location mentioned in the text, at least as close as possible. Otherwise, reading the manuscript and following the experiments could be challenging for the readers.

---

### Meta-Review · Area_Chair_VMdf · 2022-04-09

**Recommendation:** Reject
**Confidence:** 4

**Metareview:**

Reviewers highlighted that the authors wrote and released code, but remarked that it could have had better documentation. In addition, ablations would have benefitted this work.

---

### Decision · Program_Chairs · 2022-04-09

Reject